# Identifying Feedforward and Feedback Controllable Subspaces of Neural Population Dynamics

## Abstract

There is overwhelming evidence that cognition, perception, and action rely on feedback control. However, if and how neural population dynamics are amenable to different control strategies is poorly understood, in large part because machine learning methods to directly assess controllability in neural population dynamics are lacking. To address this gap, we developed a novel dimensionality reduction method, Feedback Controllability Components Analysis (FCCA), that identifies subspaces of linear dynamical systems that are most feedback controllable based on a new measure of feedback controllability. We further show that PCA identifies subspaces of linear dynamical systems that maximize a measure of feedforward controllability. As such, FCCA and PCA are data-driven methods to identify subspaces of neural population data (approximated as linear dynamical systems) that are most feedback and feedforward controllable respectively, and are thus natural contrasts for hypothesis testing. We developed new theory that proves that non-normality of underlying dynamics determines the divergence between FCCA and PCA solutions, and confirmed this in numerical simulations. Applying FCCA to diverse neural population recordings, we find that feedback controllable dynamics are geometrically distinct from PCA subspaces and are better predictors of animal behavior. Our methods provide a novel approach towards analyzing neural population dynamics from a control theoretic perspective, and indicate that feedback controllable subspaces are important for behavior.

## 1 Introduction

Feedback control has long been recognized to be central to brain function (Wiener, 1948; Conant & Ashby, 1970). Prior work has established that, at the behavioral level, motor coordination (Todorov & Jordan, 2002), speech production (Houde & Nagarajan, 2011), perception (Rao & Ballard, 1999), and navigation (Pezzulo & Cisek, 2016; Friston et al., 2012) can be accounted for by models of optimal feedback control. Advances in the ability to simultaneously record from large number of neurons have further revealed that the brain performs computations and produces behavior through low-dimensional population dynamics (Vyas et al., 2020). Together, these two facts indicate that neural population dynamics should both be able to implement the computations required to exert feedback control Friedrich et al. (2021), and be internally steerable by feedback control themselves (e.g., other brain areas controlling motor cortex to produce target dynamics). Nonetheless, methods to assess these hypotheses directly from recordings of neural population activity are absent.

The cost incurred in controlling a dynamical system is referred to as its controllability. Existing measures of controllability center around the energy (in terms of the norm of the control signal) that must be expended to steer the system state. These measures are calculated from the controllability Gramian of the (linearized) system dynamics. Controllability is an intrinsic feature of the dynamical system itself, and may be estimated from measurements of system dynamics without reference to the specific inputs to the system (Pasqualetti et al., 2013; Kashima, 2016). Network controllability analyses have delivered insights into the organization of proteomic networks (Vinayagam et al., 2016), human functional and structural brain networks (Medaglia et al., 2018; Tang & Bassett, 2018; Kim et al., 2018; Gu et al., 2015), and the connectome of *C Elegans* (Yan et al., 2017). However, prior work in network controllability has exclusively focused open loop, or feedforward, controllability in

the context of extracted networks, and not measures of closed loop, or feedback, controllability in the context of observed dynamics of data. Indeed, methods to asses feedback controllability from observations of the dynamics of neural populations are nascent.

Here, we developed dimensionality reduction methods that can be applied to neural populaiton data that maximize the feedforward and feedback controllability of extracted latent population dynamics. We first identify a correspondence between Principal Components Analysis (PCA) and the volume of state space reachable by feedforward control in linear dynamical systems (Pasqualetti et al., 2013)–this provides a control-theoretic interpretation to PCA extracted subspaces. We then present Feedback Controllable Components Analysis (FCCA), a linear dimensionality reduction method to identify feedback controllable subspaces of high dimensional dynamical systems based on a novel measure of feedback controllability.

Our focus on linear models of dynamics is a computational necessity; nonlinear measures of controllability require nonlinear systems identification and involve partial differential equations that are intractable to solve in high dimensions Scherpen (1993a); Nakamura-Zimmerer et al. (2021); Kramer et al. (2024). In contrast, a key advantage of FCCA is that it can be applied to data using only the second order statistics of the observed data itself, bypassing the need for prior system identification and making the method easily applicable to large scale neural population recordings. Furthermore, in contrast to existing approaches towards dimensionality reduction in computational neuroscience Yu et al. (2009); Pandarinath et al. (2018), *FCCA does not attempt to reconstruct the neural data with a lower dimensional subspace, but rather identifies a subspace in which dynamics optimize a functional measure* (feedback controllability). Together with a functional, control theoretic interpretation of PCA, this permits direct comparison of the neural population dynamics underlying distinct control strategies from observed neural population data.

Through theory and numerical simulations, we show that the degree of non-normality of the underlying dynamical system (Trefethen & Embree, 2020) determines the degree of divergence between PCA and FCCA solutions. In the brain, the postsynaptic effect of every neuron is constrained to be either excitatory or inhibitory by Dale's Law. This structure implies that linearized dynamics within cortical circuits are necessarily non-normal (Murphy & Miller, 2009). Prior work has highlighted the capacity of non-normal dynamical systems to retain memory of inputs (Ganguli et al., 2008) and transmit information (Baggio & Zampieri, 2021). Our results show that non-normality also plays a fundamental role in shaping the controllability of neural systems. Finally, we applied FCCA to diverse neural recordings and demonstrate that those subspaces are better predictors of behavior than PCA subspaces (despite both being linear), and that the two subspaces are geometrically distinct. This suggests that feedback controllable subspaces (FCCA) are more relevant for behavior than feedforward controllable subspaces (PCA).

## 2 CONTROLLABLE SUBSPACES OF LINEAR DYNAMICAL SYSTEMS

Here, we provide detailed derivations of our data-driven measures of controllability. We first discuss the natural cost function to measure feedforward controllabiity (eq. 4) and highlight its correspondence to PCA. Next, we present the analogous measure for feedback controllabiity (eq. 7), and how it may be estimated *implicitly* (i.e., without explicit model fitting) from the observed second order statistics of data (eq. 11). We provide rationale for this cost function as measuring the complexity of the feedback controller required to regulate the observed neural population dynamics.

We consider linear dynamical systems of the form:

$$\dot{x}(t) = Ax(t) + Bu(t) \quad y(t) = Cx(t) \tag{1}$$

where $x(t) \in \mathbb{R}^N$ is the neural state (i.e., the vector of neuronal activity, not a latent variable) and $u(t)$ is an external control input. $A \in \mathbb{R}^{N \times N}$ is the dynamics matrix encoding the effective first order dynamics between neurons. $B \in \mathbb{R}^{N \times p}$ describes how inputs drive the neural state, and $C \in \mathbb{R}^{d \times N}, d << N$ is a readout matrix projecting the neural dynamics to a lower dimensional space. The input-output behavior (i.e., the mapping from $u(t)$ to $y(t)$) can equivalently be represented in the Laplace domain using the transfer function $G(s) = C(sI - A)^{-1}B$ Kailath (1980).

Consider an invertible linear transformation of the state variable $x \to Tx$. Under such a state-space transformation, the input-output behavior of the system 1 is left unchanged as the state space matrices

transform as $(A, B, C) \to (TAT^{-1}, TB, CT^{-1})$. This implies that there are many possible choices of $(A, B, C)$ matrices, referred to as realizations, that give rise to the same transfer function $G(s)$. A minimal realization contains the fewest number of state variables (i.e., $A$ has the smallest dimension) amongst all realizations. Measures of controllabity that are *intrinsic* to the dynamical system should be invariant across all realizations. We will show that our measures of feedforward and feedback controllabillity exhibit this property.

Throughout, we will assume that the observed data obeys following underlying state dynamics:

$$\dot{x}(t) = Ax(t) + Bdw(t); \quad dw(t) \sim \mathcal{N}(0, 1); \quad y(t) = Cx(t) \tag{2}$$

Compared to eq. 1, $u(t)$ has been replaced by temporally white noise $dw(t)$, a reasonable assumption given that input signals are unmeasured in neural recordings. Our metrics of controllability rely only on observing the linear dynamics under this latent, stochastic excitation.

### 2.1 PRINCIPAL COMPONENTS ANALYSIS EIGENVALUES MEASURE FEEDFORWARD CONTROLLABILITY

A categorical definition of controllability for a dynamical system is that for any desired trajectory from initial state to final state, there exists a control signal $u(t)$ that could be applied to the system to guide it through this trajectory. For a (stable) linear dynamical system, a necessary and sufficient condition for this to hold is that the controllability Gramian, $\Pi$, has full rank. $\Pi$ is obtained from the state space parameters through the solution of the Lyapunov equation:

$$A\Pi + \Pi A^\top = -BB^\top \quad \Pi = \int_0^\infty dt\ e^{At}BB^\top e^{A^\top t} \tag{3}$$

The rank condition on $\Pi$ as a definition of controllability, while canonical (Kailath, 1980), is an all or-nothing designation; either all directions in state space can be reached by control signals, or they cannot. Furthermore, this definition does not take into account the energy required to achieve the desired transition. While certain directions in state space may in principle be reachable, the energy required to push the system in those directions may be prohibitive.

Thus, given that the system is controllable, we can ask a more refined question: what is the energetic effort required to control different directions of state space? The energy required for control is measured by the norm of the input signal $u(t)$. It can be shown (Pasqualetti et al., 2013) that to reach states that lie along the eigenvectors of $\Pi$, the minimal energy is proportional to the inverse of the corresponding eigenvalues of $\Pi$. Directions of state space that have large projections along eigenvectors of $\Pi$ with small eigenvalues are therefore harder to control. For a unit-norm input signal, the volume of reachable state space is proportional to the determinant of $\Pi$ (Summers et al., 2016).

The above intuition can be encoded into the objective function of a dimensionality reduction problem: for a fixed-norm input signal, find $C$ that maximizes the reachable volume within the subspace. This volume is measured by the determinant of $C\Pi C^\top$. Identifying subspaces of maximum feedforward controllability is then posed as the following optimization problem:

$$\operatorname{argmax}_C \log \det C\Pi C^\top \ \mid \ C \in \mathbb{R}^{d \times N}, CC^\top = I_d \tag{4}$$

Observe that under state space transformations, $\Pi$ maps to $T\Pi T^\top$, whereas $C$ maps to $CT^{-1}$. Hence, as desired, eq. 4 is invariant to state space transformations and thus an intrinsic property of the dynamical system. We include the constraint $CC^\top = I_d$ to ensure the optimization problem is well-posed. Without it, one could, for example, multiply $C$ by an overall constant and increase the objective function. We can assess this objective function from data generated by eq. 2, as in this case the observed covariance of the data will coincide with the controllability Gramian (Mitra, 1969; Kashima, 2016). The solution of problem 4 coincides with that of PCA, as the optimal $C$ of fixed dimensionality $d$ has rows given by the top $d$ eigenvectors of $\Pi$ (see Theorem 2 on pg. 7 and Lemma 1 in the Appendix).

### 2.2 LINEAR QUADRATIC GAUSSIAN SINGULAR VALUES MEASURE FEEDBACK CONTROLLABILITY

How does one quantify the feedback controllability of a system? The primary distinction between feedforward control and feedback control is that the latter utilizes observations of the state to

synthesize subsequent control signals. Feedback control therefore involves two functional stages: filtering (i.e., estimation) of the underlying dynamical state ($x(t)$) from the available observations ($y(t)$) and construction of appropriate regulation (i.e., control) signals. For a linear dynamical system, state estimation is optimally accomplished by the Kalman filter, whereas state regulation is canonically achieved via linear quadratic regulation (LQR). It will be crucial in what follows to recall that the Kalman Filter is an efficient, recursive, Gaussian minimum mean square error (MMSE) estimate of $x(t)$ given observations $y(\tau)$ for $\tau \leq t$. These two functional stages optimally solve the following cost functions:

$$\text{Kalman Filter}: \min_{p(x_0|y_{-T:0})} \lim_{T \to \infty} \text{Tr}\left(\mathbb{E}\left[(\mathbb{E}(x_0|y_{-T:0}) - x_0)(\mathbb{E}(x_0|y_{-T:0}) - x_0)^\top\right]\right)$$

$$\text{LQR}: \min_{u \in L^2[0,\infty)} \lim_{T \to \infty} \mathbb{E}\left[\frac{1}{T}\int_0^T x^\top C^\top C x + u^\top u \, dt\right]$$

where $y_{-T:0}$ denotes observations over the interval $[-T, 0]$. The minima of these cost functions are obtained from the solutions of dual Riccati equations:

$$AQ + QA^\top + BB^\top - QC^\top CQ = 0 \tag{5}$$

$$A^\top P + PA + C^\top C - PBB^\top P = 0 \tag{6}$$

where

$$Q = \min_{p(x_0|y_{-T:0})} \lim_{T \to \infty} \mathbb{E}\left[(\mathbb{E}(x_0|y_{-T:0}) - x_0)(\mathbb{E}(x_0|y_{-T:0}) - x)^\top\right]$$

$$x_0^\top P x_0 = \min_{u \in L^2[0,\infty)} \left\{\lim_{T \to \infty} \mathbb{E}\left[\frac{1}{T}\int_0^T x^\top C^\top C x + u^\top u \, dt\right], \ x(0) = x_0\right\}$$

Here, $Q$ is the covariance matrix of the estimation error, whereas $P$ encodes the regulation cost incurred for varying initial conditions ($x_0$). $\text{Tr}(P)$ is proportional to the average regulation cost over all unit norm initial conditions.

The solutions of the Riccati equations are not invariant under the invertible state transformation $x \mapsto Tx$. The filtering Riccati equation will transform as $Q \mapsto TQT^\top$ whereas $P$ will transform as $(T^{-1})^\top P T^{-1}$. As such, simply by defining new coordinates via $T$ we can shape the difficulty of filtering and regulating various directions of the state space. Therefore $Q$ and $P$ on their own are not suitable cost functions for measuring feedback controllability. However, the product $PQ$ undergoes a similarity transformation $PQ \to (T^\top)^{-1}QPT^\top$. *Hence, the eigenvalues of $PQ$ are invariant under similarity transformations, and define an intrinsic measure of the feedback controllability of a system. Additionally, there exists a particular $T$ that diagonalizes $PQ$.* Following Jonckheere & Silverman (1983), we refer to the corresponding eigenvalues as the LQG (Linear Quadratic Gaussian) singular values. In this basis, the cost of filtering each direction of the state space equals the cost of regulating it. We formalize these statements by restating Theorem 1 from Jonckheere & Silverman (1983):

**Theorem 1.** *Let $(A, B, C)$ be a minimal realization of $G(s)$. Then, the eigenvalues of $QP$ are similarity invariant. Further, these eigenvalues are real and strictly positive. If $\mu_1^2 \geq \mu_2^2 \geq \mu_N^2 > 0$ denote the eigenvalues of $QP$ in decreasing order, then there exists a state space transformation $T$, $(A, B, C) \to (TAT^{-1}, TB, CT^{-1}) \equiv (\tilde{A}, \tilde{B}, \tilde{C})$ such that:*

$$Q = P = diag(\mu_1, \mu_2, ..., \mu_N)$$

*The realization $(\tilde{A}, \tilde{B}, \tilde{C})$ will be called the closed-loop balanced realization.*

*Proof.* Let $Q = LL^\top$ be the Cholesky decomposition of $Q$ and let $L^\top PL$ have Singular Value Decomposition $U\Sigma^2 U^\top$. Then, one can check $T = \Sigma^{1/2} U^\top L^{-1}$ provides the desired transformation. $\square$

Hence, as an intrinsic measure of feedback controllability, we take the sum of the LQG singular values $\mu_i^2$, corresponding to the sum of the ensemble cost to filter and regulate each direction of the neural state space:

$$\text{Tr}(PQ) \tag{7}$$

### 2.3 THE FEEDBACK CONTROLLABILITY COMPONENTS ANALYSIS METHOD.

We developed a novel dimensionality reduction method, Feedback Controllability Components Analysis (FCCA), that can be readily applied to observed data from typical systems neuroscience experiments. To do so, we construct estimators of the LQG singular values, and hence $\text{Tr}(PQ)$, directly from the autocorrelations of the observed neural firing rates. The FCCA objective function arises from the observation that causal and acausal Kalman filtering are also related via dual Riccati equations. We first show that through an appropriate variable transformation, we obtain a state variable $x_b(t)$ whose dynamics unfold backwards in time via the same dynamics matrix ($A$) which evolves $x(t)$ (the neural state) forwards in time. Once established, this enables us to use the error covariance matrix of Kalman filtering $x_b(t)$ as a stand-in for the cost of regulating $x(t)$.

In particular, given the state space realization of the forward time stochastic linear system in eq. 2 , the joint statistics of $(x(t), y(t))$ can equivalently be parameterized by a Markov model that evolves backwards in time (L. Ljung & T. Kailath, 1976):

$$-\dot{x}_b(t) = A_b x_b(t) + B dw(t); \quad y = C x_b(t) \tag{8}$$

where $A_b = -A - BB^\top \Pi^{-1} = \Pi A^\top \Pi^{-1}$ and $\Pi = \mathbb{E}[x(t)x(t)^\top]$ is the solution of the Lyapunov equation (eq. 3)

Examination of eq. 5 and eq. 6 reveals that the filtering and LQR Riccati equations differ primarily in two respects. First, the dynamics matrix is transposed ($A \to A^\top$). Second, the inputs and outputs have been exchanged ($B \to C^\top$, $C \to B^\top$). To use the error covariance of state filtering as a stand-in for the state regulation cost, we therefore require that the corresponding acausal state dynamics (determined by $A_b$) respect these differences. To this end, consider the transformed state $x_a(t) = \Pi^{-1}x(t)$. Substituting $x(t) = \Pi x_a(t)$ and $A_b = \Pi A^\top \Pi^{-1}$ into the equations for the backward dynamics result in following dynamics for this *adjoint* state:

$$-\dot{x}_a(t) = A^\top x_a(t) + \Pi^{-1}B dw(t)$$

Then, if we construct a readout of this transformed state $y_a(t) = C\Pi x_a(t) = Cx(t)$, the Riccati equation associated with Kalman filtering $x_a$, whose solution we denote $\tilde{P}$, takes on the form:

$$A^\top \tilde{P} + \tilde{P}A + \Pi^{-1}BB^\top \Pi^{-1} - \tilde{P}\Pi C^\top C \Pi \tilde{P} = 0 \tag{9}$$

$$A^\top P + PA + C^\top C - PBB^\top P = 0 \tag{eq 6}$$

We see that eq. 9 coincides with eq. 6 (reproduced for convenience) upon switching the inputs and outputs ($B \to C^\top$, $C \to B^\top$) and reweighting them by a factor of $\Pi^{-1}$ and $\Pi$, respectively. In fact, eq. 9 coincides with the Riccati equation associated with a slightly modified LQR problem:

$$\min_{u \in L^2[0,\infty)} \quad \lim_{T \to \infty} \mathbb{E}\left[\frac{1}{T}\int_0^T x^\top \Pi^{-1}BB^\top \Pi^{-1}x + u^\top \Pi^2 u \, dt\right] \tag{10}$$

This is the regulator problem for the adjoint state $x_a(t) = \Pi^{-1}x(t)$. Therefore, under the assumption that the observed dynamics can be approximated by a linear dynamical system, we can measure LQG singular values associated with this modified LQR problem directly from measuring the causal minimum mean square error (MMSE) associated with prediction of $x(t)$ ($Q$), and the acausal MMSE associated with prediction of $x_a(t)$ ($\tilde{P}$).

To explicitly construct an estimator of the quantity $\text{Tr}(\tilde{P}Q) = \text{Tr}(Q\tilde{P})$, recall the matrix $Q$ is the error covariance of MMSE prediction of the system state $x(t)$ given past observations $y(t)$ over the interval $(t - T, t)$, whereas the matrix $\tilde{P}$ is the error covariance of MMSE prediction of the transformed system state $x_a(t)$ given future observations $y_a(t)$ over the interval $(t, t + T)$. The choice of $T$ is the only hyperparameter associated with FCCA. As discussed above, the Kalman Filter is used to efficiently calculate these MMSE estimates given an explicit state space model of the dynamics. In our case, to keep system dynamics implicit, we instead directly use the formulas for the MMSE error covariance in terms of cross correlations between $x(t), x_a(t)$ and $y(t), y_a(t)$. The standard formulas for the error covariance of MMSE prediction of a Gaussian distributed variable $z$ given $v$ read: $\Sigma_z - \Sigma_{zv}\Sigma_v^{-1}\Sigma_{vz}^\top$ where $\Sigma_z = \mathbb{E}[zz^\top], \Sigma_v = \mathbb{E}[vv^\top]$ and $\Sigma_{zv} = \mathbb{E}[zv^\top]$. The FCCA

objective function is thus:

$$
\text{FCCA:} \quad \operatorname{argmin}_C \operatorname{Tr}\left[ \underbrace{\left(\Pi - \Lambda_{1:T}(C)\Sigma_T^{-1}(C)\Lambda_{1:T}^\top(C)\right)}_{\text{causal MMSE covariance } (Q)} \underbrace{\left(\Pi^{-1} - \tilde{\Lambda}_{1:T}^\top(C)\Sigma_T^{-1}(C)\tilde{\Lambda}_{1:T}(C)\right)}_{\text{acausal MMSE covariance } (\tilde{P})} \right]
$$
(11)

where for discretization timescale $\tau$,

$$
\Pi = \underset{\text{(covariance of the neural data)}}{\mathbb{E}[x(t)x(t)^\top]} , \ \Lambda_k = \underset{\text{(autocorrelation of the neural data)}}{\mathbb{E}[x(t+k\tau)x(t)^\top]} , \ \tilde{\Lambda}_k = \underset{\text{(autocorrelations of the adjoint state)}}{\mathbb{E}[x_a(t+k\tau)x_a(t)^\top]}
$$

$$
\Lambda_{1:T}(C) = \{\Lambda_1 C^\top, \Lambda_2 C^\top, ..., \Lambda_T C^\top\}, \ \tilde{\Lambda}_{1:T}(C) = \{\tilde{\Lambda}_1 \Pi C^\top, \tilde{\Lambda}_2 \Pi C^\top, ..., \tilde{\Lambda}_T \Pi C^\top\}
$$

and $\Sigma_T(C)$ is a block-Toeplitz space by time covariance matrix of $y(t)$ (i.e. the $ij^{\text{th}}$ block of $\Sigma_T(C)$ is given by $C^\top \Lambda_{|i-j|} C$. We optimize the FCCA objective function via L-BFGS.

## 2.4 CONTROL-THEORETIC INTUITION FOR FCCA

We have shown how the sum of LQG singular values is an intrinsic measure of the cost to filter/regulate a linear dynamical system which is minimized at a fixed readout dimensionality by FCCA. We now provide further intuition for FCCA. In order to control the system state and carry out the computations necessary to perform state estimation and control signal synthesis, the controller itself must implement its own internal state dynamics. Thus, in addition to the complexity of the system itself, we may inquire about the complexity of the controller. One intuitive measure of this complexity is given by the controller's state dimension (i.e., the McMillan degree), or the number of dynamical degrees of freedom it must implement to function. In the context of brain circuits, the degrees of freedom of the controller must ultimately be implemented via networks of neurons. We therefore hypothesize that biology favors performing task relevant computations via dynamics that require low dimensional controllers to regulate. As we argue below, minimizing the sum of LQG singular values over readout matrices ($C$) corresponds to a relaxation of the objective of searching for a subspace that enables control via a controller of low dimension. In other words, FCCA searches for dynamics that can be regulated with controllers of low complexity.

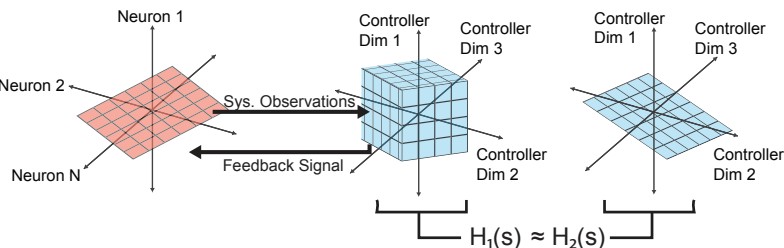

**Figure 1:** In principle, a controller of dimension as large as the neural state space may be required to effectively regulate dynamics within a FBC subspace ($H_1(s)$). However, subspaces optimized to minimize either the rank, or more practically, the trace of $PQ$ will require controllers of lower dimensionality to achieve near-optimal performance ($H_2(s)$).

Recall from Theorem 1 above that there exists a linear transformation that simultaneously diagonalizes both $P$ and $Q$. Let $(\tilde{A}, \tilde{B}, \tilde{C})$ be the corresponding balanced realization. Order the LQG singular values in descending magnitude $\{\mu_1, ..., \mu_N\}$ and divide them into two sets $\{\mu_1, ..., \mu_m\}$ and $\{\mu_{m+1}, ..., \mu_N\}$. Assume the system input is of dimensionality $p$ and the output is of dimension $d$ (i.e., $\tilde{B} \in \mathbb{R}^{N \times p}$ and $\tilde{C} \in \mathbb{R}^{d \times N}$). Then, one can partition the state matrices $\{\tilde{A}, \tilde{B}, \tilde{C}\}$ accordingly:

$$
\tilde{A} = \begin{bmatrix} A_{11} & A_{12} \\ A_{21} & A_{22} \end{bmatrix} \quad \tilde{B} = \begin{bmatrix} B_1 \\ B_2 \end{bmatrix} \quad \tilde{C} = [C_1 \quad C_2]
$$

Where $A_{11} \in \mathbb{R}^{m \times m}, A_{22} \in \mathbb{R}^{N-m \times N-m}, B_1 \in \mathbb{R}^{m \times p}, B_2 \in \mathbb{R}^{N-m \times p}, C_1 \in \mathbb{R}^{d \times m}, C_2 \in \mathbb{R}^{d \times N-m}$. It can be shown that the optimal controller of dimension $m$ is obtained from solving the Riccati equations corresponding to the truncated system $(A_{11}, B_1, C_1)$. If the LQG singular values $\{\mu_{m+1}, ..., \mu_N\}$ are negligible, then the controller dimension can be reduced with essentially no loss in regulation performance. We illustrate this idea schematically in **Figure 1**, where the controller with transfer function $H_1(s)$ is approximated by a controller with lower state dimension $H_2(s)$. This suggests that to search for subspaces of neural dynamics that require low dimensional controllers to regulate, one should minimize the objective function $\text{argmin}_C \text{Rank}(\tilde{P}Q)$, where $\tilde{P}$ and $Q$ are the solutions to the Riccati equations 9 and 5, respectively. However, rank minimization is an NP-hard problem. A convex relaxation of the rank function is the nuclear norm (i.e. the sum of the singular values) (Fazel et al., 2004). Given that $\tilde{P}Q$ is a positive semi-definite matrix, a tractable objective function that seeks subspaces of dynamics that require low complexity controllers is given by:

$$\text{argmin}_C \text{Tr}(\tilde{P}Q)$$

which is precisely what FCCA minimizes in a data-driven fashion (eq. 11).

## 3 PCA AND FCCA SUBSPACES DIVERGE IN NON-NORMAL DYNAMICAL SYSTEMS

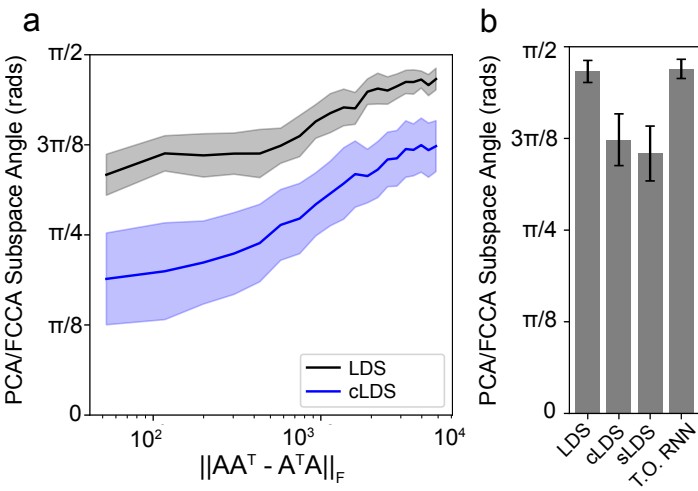

**Figure 2:** (a) Average subspace angles as a function of non-normality between $d = 2$ FCCA and PCA projections of rate activity from Dale's law constrained linear dynamical systems (LDS, Black) and from firing rates derived from spiking activity driven by Dale's Law constrained LDS (count LDS, Blue). (b) Average subspace angles between FCCA and PCA projections at the highest value of non-normality considered within LDS, count LDS (cLDS), switching LDS (sLDS), and task optimized RNNs (T.O. RNN). Spread and errorbars indicate standard deviations over random generations of A matrices and 10 random initializations of FCCA.

Having derived data driven optimization problems to identify feedforward (PCA) and feedback (FCCA) controllable subspaces, we investigated under what conditions the solutions of PCA and FCCA will be distinct. We found that a key feature of the dynamical system of eq. 1 that determines the similarity of PCA and FCCA solutions is the non-normality of the underlying dynamics matrix, $A$. We first prove that when $A$ is normal (symmetric), and $B = I$, the critical points of PCA (eq. 11) and the FCCA objective function (eq. 7) coincide. [1]

**Theorem 2.** *For $B = I_N, A = A^\top, A \in \mathbb{R}^{N \times N}$, with all eigenvalues of $A$ distinct and $\max Re(\lambda(A)) < 0$, the critical points of the feedforward controllability objective function eq. 4 and the feedback controllability objective function eq. 7 for projection dimension $d$ coincides with the eigenspace spanned by the $d$ eigenvalues with largest real value.*

---

[1]The set of real-valued, normal $A$ matrices can be divided into symmetric and orthogonal matrices. We restrict our treatment to stable dynamical systems. As orthogonal matrices give rise to systems that are only marginally stable, below we will use normal $A$ to refer interchangeably to symmetric $A$.

The proof of the theorem is provided in the Appendix. The restriction to $B = I$ is made within the proof, but does not apply to the general application of the method. Intuitively, in the case of symmetric, stable $A$, perturbations exponentially decay in all directions, and so the maximum response variance, and hence greatest feedforward controllability, is contained in the subspace with slowest decay, which corresponds to the eigenspace spanned by the $d$ eigenvalues with largest real value. The intuition for the slow eigenspace of $A$ serving as a (locally) optimal projection in the feedback controllability case is given by the fact that state reconstruction from past observations, the goal of the Kalman filter, will occur optimally using observations that have maximal autocorrelations with future state dynamics. Similarly, for the LQR, for a fixed rank input, the most variance will be suppressed by regulating within the subspace with slowest relaxation dynamics.

Importantly, due to Dale's Law, brain dynamics are generated by non-normal dynamical systems. To demonstrate the effect of increasing the non-normality of $A$ on the solutions of PCA and FCCA, we turn to numerical simulations (the optimal feedback controllable projections are not analytically tractable). We generated 200-dimensional dynamics matrices constrained to follow Dale's Law with an equal number of excitatory and inhibitory neurons. Neurons were connected randomly with a uniform connection probability of 0.25. To tune the non-normality of the system, we vary the strength of synaptic weights in the neuronal connectivity matrix. The strength of synaptic weights determines the spectral radius of the corresponding matrices (Rajan & Abbott, 2006). Leaving the excitatory weights fixed, we then optimize the inhibitory weights as detailed in (Hennequin et al., 2014) to ensure system stability. The resulting matrices will have enhanced non-normality, with the degree of resulting non-normality having, empirically, a monotonic relationship with the starting spectral radius. We applied our methods both directly to the cross-covariance matrices of the resulting linear dynamical systems, as well as to spiking activity driven by simulated $x_t$. In the latter case, spiking activity was generated as a Poisson process with rate $\lambda_t = \exp(x_t)$. Firing rates were obtained by binning spikes and applying a Gaussianizing boxcox transformation (Sakia, 1992). These rates were then used to estimate the cross-covariance matrices. This procedure mirrors that which was applied to neural data in the subsequent section.

In **Figure 2a**, we plot the average subspace angles between FCCA and PCA for $d = 2$ projections (other choices of $d$ shown in **Figure A1**) applied both directly to cross-covariance matrices of the linear dynamical systems (LDS, black) and cross-covariance matrices estimated from spiking activity (Count LDS, blue) as a function of the non-normality of the underlying $A$ matrix (measured using the Henrici metric, $||A^\top A - AA^\top||_F$). In both cases, we observe a nearly monotonic increase in the angles between FCCA and PCA subspaces as non-normality is increased. We note that as we constrain $A$ matrices to follow Dale's Law, we cannot tune them to be completely normal, and hence the subspace angles between FCCA and PCA remain bounded away from zero even at the lower end of non-normality. We verified that the large subspace angles between FCCA and PCA also persist in more general data generation processes. We considered non-stationary dynamics arising from a sequence of switched non-normal linear dynamical systems, and nonlinear dynamics arising from an RNN obeying Dale's Law trained to reproduce muscle EMG activity in response to a low dimensional "go cue" input signal Sussillo et al. (2015). Full details of model construction and training are provided in the Appendix. In **Figure 2b**, we plot the average FCCA/PCA subspace angles at the highest degree of model non-normality for each synthetic system (full results across all levels of non-normality are provided in **Figure A2**). In all cases, FCCA and PCA subspaces are geometrically distinct. Given the generality of non-normal dynamics due to Dale's Law, this new control-theoretic result suggests that PCA and FCCA subspaces should also be geometrically distinct in neural population data.

## 4 FCCA SUBSPACES ARE BETTER PREDICTORS OF BEHAVIOR THAN PCA SUBSPACES

We first applied FCCA to neural population recordings from the rat hippocampus made during a maze navigation task. Further details on the dataset and preprocessing steps used are provided in the Appendix. In each recording session, we fit PCA and FCCA to neural activity across a range of projection dimensions. In line with the predictions of our theory and numerical simulations, we find that the subspace angle between PCA and FCCA was consistently large across recording sessions ($> 3\pi/8$, **Figure 3a**, median and IQR indicated). We used $T = 3$ (time bins) as the FCCA hyperparameter. As FCCA is a nonconvex optimization problem, we initialized optimization

**Table 1:** FCCA/PCA comparison across neural datasets

| Dataset/Brain Region | $N_r$ | $\theta$ (deg.) | Peak Percent $\Delta\text{-}r^2$ | $\Delta\text{-}r^2$ AUC |
|---|---|---|---|---|
| Hippocampus | 8 | $74.6 \pm 1.7$ | $465 \pm 144\%$ | $3.14 \pm 0.30$ |
| M1 random | 35 | $58.0 \pm 1.1$ | $229 \pm 58\%$ | $2.75 \pm 0.12$ |
| S1 random | 8 | $67.5 \pm 3.8$ | $761 \pm 189\%$ | $2.47 \pm 0.36$ |
| M1 maze | 5 | $49.4 \pm 4.3$ | $290 \pm 72\%$ | $1.45 \pm 0.23$ |

from many random orthogonal projection matrices and choose the final solution that yields the lowest value of the cost function 11. In **Supplementary Figure A3**, we confirm that the substantial subspace angles between FCCA and PCA are largely insensitive to the choice of T, the choice of projection dimensionality, and robust across initializations of FCCA. Thus, we find that feedforward and feedback controllable subspaces are geometrically distinct in neural activity.

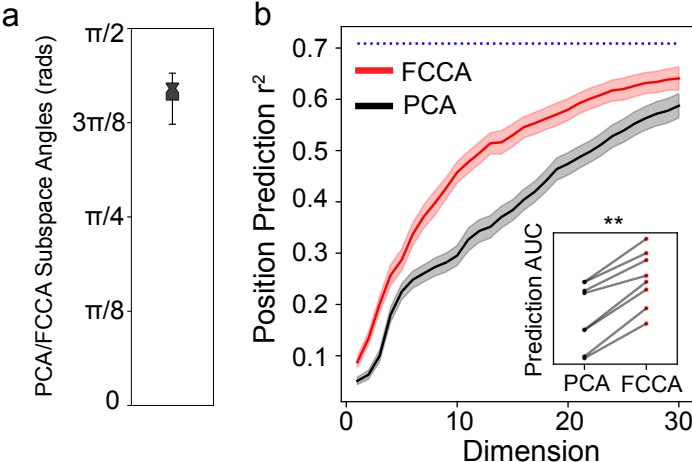

**Figure 3:** (a) Average subspace angles between FCCA and PCA at $d = 2$ across recording sessions (median $\pm IQR$ indicated). (b) Five-fold cross-validated position prediction $r^2$ as a function of projection dimension between for FCCA (red) and PCA (black) and without dimensionality reduction (dashed blue). Mean $\pm$ standard error across folds and recording sessions indicated. (inset) Total area under the curve (AUC) of decoding performance averaged over folds for PCA and FCCA within each recording session (** : $p < 10^{-2}, n = 8$, Wilcoxon signed rank test)

We next assessed the extent to which feedback controllable dynamics (as identified by FCCA), as opposed to feedforward controllable dynamics (as identified by PCA) were relevant for behavior. We trained linear decoders of the rat position from activity projected into FCCA and PCA subspaces. We used a window of 300 ms of neural activity centered around each time point to predict the corresponding binned position variable. We used linear decoders to emphasize the structure in the different subspaces available to a simple read-out. In **Figure 3b**, we report five-fold cross-validated prediction accuracy for PCA (black) and FCCA (red) over a range of projection dimensions (mean $\pm$ standard error across recording sessions and folds indicated). We found activity within FCCA subspaces to be more predictive of behavior than PCA subspaces across all dimensions, with a peak improvement of 112% at $d = 13$. This superior decoding performance additionally held consistently across each recording session individually. In the inset of **Fig. 3c**, we plot the total area under prediction $r^2$ curves shown for each recording session (FCCA significantly higher than PCA, **: $p < 10^{-2}, n = 8$, Wilcoxon signed rank test). In **Figure A4**, we verify that the superior decoding performance of FCCA subspaces hold consistently across each individual initialization. Feedback controllable subspaces therefore better capture behaviorally relevant dynamics than feedforward controllable subspaces.

To validate the robustness of these results, we repeated our analyses in two other datasets: recordings from macaque primary motor (M1 random) and primary somatosensory (S1 random) cortices during

a self paced reaching task (O'Doherty et al. (2018)), and recordings from macaque primary motor cortex during a delayed reaching task (M1 maze, Churchland et al. (2012)). Further details on data preprocessing are provided in the Appendix. In **Table 1**, we report the number of recording sessions ($N_r$), mean: average subspace angle between FCCA and PCA subspaces at $d = 2$ ($\theta$), peak percent $\Delta$-$r^2$ of behavioral prediction, and difference in the area under the behavioral prediction curves between PCA and FCCA. In all cases, standard errors are taken across the recording sessions, and analogously to **Figure 3**, behavioral decoding was performed from $d = 1$ to $d = 30$. Importantly, in all datasets, FCCA performed better behavioral prediction, and the subspace angles between FCCA and PCA were substantially different from zero.

## 5 DISCUSSION

We developed FCCA, a novel dimensionality reduction method that identifies feedback controllable subspaces of neural population dynamics. Further, the correspondence between PCA and feedforward controllability, long known in the control theory community (Moore, 1981), but unrecognized in the neuroscience community, adds additional interpretative value to these subspaces. Importantly, to the best of our knowledge, FCCA is the first method to encode functional measures of dynamics (in this case, controllability) into the objective of a dimensionality reduction method. As such, it is not designed to optimally reconstruct the neural data or maximize behavioral decoding, but rather to provide insight into the specific computations different components of neural activity are optimized for. This renders it distinct from prior latent variable analysis methods in neuroscience (e.g., GPFA Yu et al. (2009), LFADS Pandarinath et al. (2018)), and motivates the development of other methods for neural data analysis that reduce neural activity on the basis of normative, functional measures.

We demonstrated that feedforward and feedback controllable subspaces are geometrically distinct in non-normal dynamical systems, a fact of fundamental importance to the analysis of neural dynamics from cortex, where Dale's Law necessitates non-normality. Correspondingly, in electrophysiology recordings from across the brain, we found large subspace angles between FCCA and PCA subspaces. Furthermore, we found that FCCA subspaces were better predictors of behavior than PCA subspaces. This suggests that targeting feedback controllable subspaces in the design of brain machine interfaces may be advantageous in terms of accuracy of behavioral prediction, the number of samples needed to calibrate predictions to a desired level of accuracy, and the efficacy of closed loop perturbations.

Several methodological extensions to FCCA are possible. While performing dimensionality reduction on the basis of nonlinear measures of controllability remains computationally infeasible due to the need to solve high dimensional PDEs within the inner optimization loop ((Scherpen, 1993b)), FCCA could be augmented with a nonlinear encoder. In FCCA, we rely on estimation of the regulator cost through acausal filtering (eq. 9 and estimate the filtering error through the Gaussian MMSE formula (eq. 11) to keep the model of the data implicit. These correspondences only hold for linear systems under a particular choice of the LQR cost function (eq. 10). While this makes the method computationally efficient, it restricts the form of weight matrices in the LQR objective functions that can be considered. The objective function in eq. 7 could alternatively be applied to *post-hoc* analysis of linear state space models fit to neural recordings (Gao et al., 2015), as these models explicitly yield the system matrices required to solve the Riccati equations 5 and 6. This analysis could be combined with techniques from inverse linear optimal control (Priess et al., 2014) to provide a more refined picture of the controllability of population dynamics.

## 6 REPRODUCIBILITY STATEMENT

The codebase associated with the FCCA method and that used to perform numerical experiments in Section 3 and to obtain the results associated with **Figure 3** in hippocampal data have been included as supplementary materials with the submission. Intermediate data files associated with these experiments, as well as the spike sorted hippocampal dataset associated with **Figure 3** have been uploaded anonymously to figshare (instructions contained in the associated supplementary materials). The M1/S1 random dataset is publicly available at `https://zenodo.org/records/583331`, while the M1 dataset is publicly available at `https://dandiarchive.org/dandiset/000070?search=churchland&pos=1`. Full detils of the pre-processing applied to neural datasets is presented in the Appendix section A.2. A full proof of Theorem 2 is provided in the Appendix section A.3.

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

## A  APPENDIX

### A.1  DETAILS OF SWITCHING LDS AND RNN TRAINING

For results associated with the switching LDS (sLDS, (Linderman et al., 2016)) in **Figure 2b**, we simulated data from a system that switched between a sequence of three $A$ matrices (still constrained to follow Dale's law) in eq. 2 with roughly equivalent degree of non-normality.

The task optimized RNN (T.O. RNN) was comprised of 300 hidden units with ReLU nonlinearities. The recurrent connectivity was initialized in the same manner as the LDS and count LDS systems

described in the results associated with **Figure 2**. Thus, these networks had sparse connectivity, and were constrained to follow Dale's Law. We enforced Dale's law and the initial sparsity pattern throughout network training. The RNNs were trained to produce muscle electromyography (EMG) activity recorded from a macaque monkey performing a reaching task, as described in Churchland et al. (2012); Sussillo et al. (2015). Briefly, the dataset consisted of 216 unique task conditions and an 8 dimensional target EMG time series for each condition. Following Sussillo et al. (2015), the RNNs were provided with a sixteen dimensional square wave pulse input to represent an experimental "go" cue. We optimized the RNN input matrix, output matrix, weight matrix, and input and state biases using Adam over five different initializations of the weight ($A$) matrix. The trained networks exhibited a close fit to the target EMG activity ($r^2 = 0.99$). To fit FCCA and PCA, we concatenated the time series of hidden activations across all conditions together, mirroring the structure of the M1/S1 random dataset.

## A.2 Details of neural datasets

Data from the hippocampus contained recordings from a single rodent. There were a total of 8 recording sessions lasting approximately 20 minutes each with between 98-120 identified single units within each recording session. We performed our analyses on neural activity while the rat was in motion (velocity $> 4$ cm/s).

The M1/S1 random dataset contained a total of 35 recording sessions from 2 monkeys (28 within monkey 1, 7 within monkey 2) spanning 17309 total reaches (13149 from monkey 1, 4160 from monkey 2). Of the 35 recording sessions, 8 included activity from S1. The number of single units in each recording session varied between 96-200 units in M1, and 86-187 in S1. The maze dataset contained 5 recording sessions recorded from 2 different monkeys comprising 10829 total reaches (8682 in monkey 3, 2147 in monkey 4). Each recording session contained 96 single units. Both datasets mapped the monkey hand location to a cursor location on the 2D task plane. For the M1/S1 random dataset, we decoded cursor velocity, whereas for the maze dataset, we decoded cursor position.

We binned spikes within the hippocampal data at 25 ms, and the M1/S1 random and M1 maze datasets at 50 ms. We then applied a boxcox transformation to binned firing rates to Gaussianize the data. A single fit of FCCA on the activity from a single recording session in the datasets considered using a desktop computer equipped with an 8 core CPU and 64 GB of memory requires $< 5$ seconds.

## A.3 Proof of Theorem 2

In this section, we prove the equivalence of the solutions of the FFC (eq. 4) and FBC objective functions (eq. 7) when system dynamics are stable and symmetric. We focus on symmetric matrices as the requirement that dynamics be stable (i.e., all eigenvalues of the dynamics $A$ must have negative real part) essentially reduces the space of normal matrices to that of symmetric matrices. We reproduce these objective functions for convenience:

$$C_{\text{FFC}} : \quad \text{argmax}_C \log \det C\Pi C^\top$$
$$C_{\text{FBC}} : \quad \text{argmin}_C \text{Tr}(PQ)$$

We prove this theorem when the matrix $P$ in the FBC objective function arises from the canonical LQR loss function:

$$\min_u \left\{ \lim_{T \to \infty} \mathbb{E} \left[ \frac{1}{T} \int_0^T x^\top x + u^\top u \, dt \right], \quad x(0) = x_0, u \in L^2[0, \infty) \right\}$$

and not the variant given in eq. 10. When calculating FBC from data within FCCA, we must use the latter LQR loss function as it maps onto acausal filtering, and therefore may be estimated from data. Recall from the discussion below eq. 8 that within the FFC objective function, we assess controllability when the output/observation matrix $C$ is used as the input matrix for the regulator signal (i.e., we make the relabeling $B^\top \to C$. We further work under the assumption that the input

matrix $B$ to the open loop system is equal to the identity. The open loop dynamics of $x(t)$ are then given by:

$$\dot{x} = Ax(t) + u(t) \tag{12}$$

where $u(t)$ has the same dimensionality as $x(t)$, and is uncorrelated with the past of $x(t)$ (i.e. $u(t) \perp x(\tau), \tau < t$). Formally, $u(t)$ represents the innovations process of $x(t)$. The equations for $Q$ (corresponding to the Kalman Filter, eq. 5) and the equation for $P$ (corresponding to the LQR, eq. 6) reduce to the following:

$$AQ + QA + I_N - QC^\top CQ = 0 \tag{13}$$

$$AP + PA + I_N - PCC^\top P = 0 \tag{14}$$

where $I_N$ denotes the $N \times N$ identity matrix.

We observe that under the stated assumptions, the Riccati equations for $Q$ and $P$ actually coincide, and thus the FBC objective function reads $Tr(Q^2)$. We will show that both FFC and FBC objective functions achieve local optima for some fixed projection dimension $d$ when the projection matrix $C$ coincides with a projection onto the eigenspace spanned by the $d$ eigenvalues of A with largest real part, which we denote as $V_d$. In fact, in the case of the FFC objective function, the eigenspace corresponds to a global optimum. For the FBC objective function, we are able to establish global optimality rigorously only for the $2D \rightarrow 1D$ dimension reduction.

We briefly outline the proof strategy. First, we will prove the optimality of $V_d$ for the FFC objective function in section S1.9.1 by showing that (i) $V_d$ is an eigenvector of $\Pi$ in the case when $A$ is symmetric and (ii) relying on the Poincare Separation Theorem. Then, in section S1.9.2, we will prove that $V_d$ is a critical point of the FBC objective function. The proof relies on an iterative technique to solve the Riccati equation. These iterates form a recursively defined sequence that provide increasingly more accurate approximations to the FBC objective function that converge in the limit. Treating these iterative approximations of the FBC objective function as a function of $C$, we show that $V_d$ is a critical point of all iterates, and thus in the limit, $V_d$ is a critical point of the FBC objective function.

**FFC Objective Function**

**Lemma 1.** *For $B = I_N, A = A^\top, A \in \mathbb{R}^{N \times N}$, with all eigenvalues of $A$ distinct and $\max Re(\lambda(A)) < 0$, the optimal solution for the feedforward controllability objective function for projection dimension $d$ coincides with $V_d$, the matrix whose rows are formed by the eigenvectors corresponding to the $d$ eigenvalues of $A$ with largest real value.*

*Proof*

The FFC objective function reads:

$$\operatorname{argmax}_C \log \det C\Pi C^\top \mid C \in \mathbb{R}^{d \times N}, CC^\top = I_d \tag{15}$$

We first re-write $\Pi$:

$$\Pi = \int_0^\infty dt\, e^{At} BB^\top e^{A^\top t} = \int_0^\infty dt\, e^{2At}$$

Let $A = U\Lambda U^\top$ denote the eigenvalue decomposition of $A$. Recall that since $A = A^\top$, $U$ is orthogonal. Then we can write:

$$\Pi = U \int_0^\infty dt e^{2\Lambda t} U^\top$$

$$= \frac{1}{2} UDU^\top$$

where $D$ is a diagonal matrix with diagonal entries $\{\frac{1}{-\lambda_1}, \frac{1}{-\lambda_2}, ..., \frac{1}{-\lambda_N}\}$ being the eigenvalues of $\Pi$. We conclude that the matrix $\Pi$ has the same eigenbasis as $A$. Also, since all $\lambda_j$ are real and negative, the ordering of the eigenvalues is preserved ($\lambda_i > \lambda_j$ implies $-\frac{1}{\lambda_i} > -\frac{1}{\lambda_j}$).

That $V_d$ solves 15 follows from the Poincare separation theorem, which we restate for convenience:

**Proposition 1.** *Poincare Separation Theorem (Magnus & Neudecker (2019), 11.10)*

*Let $M$ be any square, symmetric matrix, and let $\mu_1 \geq \mu_2 \geq \ldots \geq \mu_N$ be its eigenvalues. Let $C \in \mathbb{R}^{d \times N}$ be a semi-orthogonal matrix (i.e., $CC^\top = I_d$). Then, the eigenvalues $\eta_1 \geq \eta_2 \geq \ldots \geq \eta_d$ of $CMC^\top$ satisfy:*

$$\mu_i \geq \eta_i \geq \mu_{N-d+i}$$

In particular, Proposition 1 implies that $\det CMC^\top = \prod_{i=1}^d \eta_i \leq \prod_{i=1}^d \mu_i$, and hence $\log \det CMC^\top \leq \sum_{i=1}^d \log \mu_i$ We now show that this inequality is satisfied with equality when $C = V_d$. Consider the optimization problem

$$\text{argmax}_C \log \det CMC^\top \mid C \in \mathbb{R}^{d \times N}, CC^\top = I_d \tag{16}$$

Let $M = U\Gamma U^\top$ be the eigendecomposition of $M$. We can equivalently parameterize the optimization problem as:

$$\text{argmax}_{\tilde{C}} \log \det \tilde{C}\Gamma\tilde{C}^\top \mid \tilde{C} \in \mathbb{R}^{d \times N}, \tilde{C}\tilde{C}^\top = I_d \tag{17}$$

The solution to the original problem, eq. 16, can be recovered from setting $C = \tilde{C}U^\top$. Now, assume (without loss of generality) that we have arranged the values of $\Gamma$ so that the largest d eigenvalues, $\mu_1, ..., \mu_d$, occur first. We observe that the choice of $\tilde{C} = [I_d; \mathbf{0}_{N-d,N-d}] \equiv \tilde{C}_*$, which picks out these first d elements of the diagonal of $\Gamma$, yields $\log \det \tilde{C}_*^\top \Gamma \tilde{C}_* = \sum_{i=1}^d \log \mu_i$, and hence solves the desired optimization problem. It follows that $C_* = \tilde{C}_* U^\top = V_d$

To complete the proof of Lemma 1, we substitute $M$ with $\Pi$, and the eigenvalues $\mu_i$ with $-1/\lambda_i$ (the eigenvalues of $\Pi$, expressed in terms of the eigenvalues of $A$). $\square$

**FBC Objective Function**

For the case of the FBC objective function, we show that projection matrices of rank $d$ that align with the $d$ slowest eigenmodes of $A$ constitute local minima of the objective function. We rely on two simplifying features of the problem. First, the FBC objective function is invariant to the choice of basis in the state space. We therefore work within the eigenbasis of $A$, as within this basis, the system defined by eq. 12 decouples into $n$ non-interacting scalar dynamical systems. Additionally, we rely on the fact that the FBC objective function is also invariant to coordinate transformations within the projected space. In other words, the choice of coordinates in which we express $y$ also makes no difference. Without loss of generality then, we may treat the problem in a basis where $A$ is diagonal with entries given by its eigenvalues and $C$ is an orthonormal projection matrix (i.e. $CC^\top = I_d$). A restatement of the latter condition is that $C$ belongs to the Steifel manifold of $N \times d$ matrices: $\Omega \equiv \{C \in \mathbb{R}^{N \times d} | CC^\top = I_d\}$.

**Lemma 2.** *For $B = I_N, A = A^\top, A^{N \times N}$, with all eigenvalues of $A$ distinct and $\max Re(\lambda(A)) < 0$, the projection matrix onto the eigenspace spanned by the $d$ eigenvalues of $A$ with largest real value constitutes a critical point of the LQG trace objective function on $\Omega$*

*Proof* Explicitly calculating the gradient of the solution of the Riccati equation is analytically intractable for $n > 1$, and so we we will rely on the analysis of an iterative procedure to solve the Riccati equation via Newton's method, known as the Newton-Kleinmann (NK) iterations (Kleinman, 1968). These iterations are described in the following proposition:

**Proposition 2.** *Consider the Riccati equation $0 = AQ + QA^\top + BB^\top - QC^\top CQ$. Let $Q_m, m = 1, 2, ...$ be the unique positive definite solution of the Lyapunov equation:*

$$0 = A_k Q_m + Q_m A_k^\top + BB^\top + Q_{m-1} C^\top C Q_{m-1} \qquad (18)$$

*where $A_k = A - C^\top C Q_{m-1}$, and where $Q_0$ is chosen such that $A_1$ is a stable matrix (i.e. all real parts of its eigenvalues are $< 0$). For two positive semidefinite matrices $M, N$, we denote $M \geq N$ if the difference $M - N$ remains positie semidefinite. Then:*

1. *$Q \leq Q_{m+1} \leq Q_m \leq ..., k = 0, 1$*

2. *$\lim_{k \to \infty} Q_m = Q$*

Thus the $Q_m$ iteratively approach the solution of the Riccati equation from above. Since in our case, the Riccati equations for $P$ and $Q$ coincide, an identical sequence $P_k$ can be constructed using analogous NK iterations that approaches $P$ from above. From this, it follows that $\lim_{k \to \infty} Tr(Q_m P_k) = \lim_{k \to \infty} Tr(Q_m^2) = Tr(Q^2)$. We then use the fact that in addition to the $Q_m$ converging to $Q$, the sequence $\nabla_C \text{Tr} \left( Q_m^2 \right)$ converges to $\nabla_C \text{Tr}(Q^2)$ as $k \to \infty$, where $\nabla_C$ denotes the gradient with respect to $C$. This is rigorously established in the following lemma, which is the multivariate generalization of Theorem 7.17 from (Rudin & others, 1976):

**Lemma 3.** *Suppose $\{f_m\}$ is a sequence of functions differentiable on an interval $h \subset \mathcal{H}$, where $\mathcal{H}$ is some finite-dimensional vector space, such that $\{f_m(x_0)\}$ converges for some point $x_0 \in h$. If $\{\nabla f_m(x_0)\}$ converges uniformly in $h$, then $\{f_m\}$ converges uniformly on $I$, to a function $f$, and*

$$\nabla f(x) = \lim_{m \to \infty} \nabla f_m(x) \quad x \in h$$

Here, the $\{f_m\}$ are the Newton-Kleinmann iterates $Q_m$, and $x_0$ corresponds to the $C$ matrix that projects onto the slow eigenspace of $A$. The NK iterates are known to converge uniformly over an interval of possible $C$ matrices (in fact any such $C$ matrix for which there exists a $K$ such that $A - C^T C K$ is a stable matrix) (Kleinman, 1968).

We will calculate the gradient $\nabla_C Q_m$ on $\Omega$ by explicitly calculating the directional derivatives of $Q_m$ over a basis of the tangent space of $\Omega$ at $C_{\text{slow}}$. Any element $\Psi$ belonging to the tangent space at $C \in \Omega$ can be parameterized by the following (Edelman et al., 1998):

$$\Psi = CM + (I_N - CC^\top)T$$

where $M$ is skew symmetric and $t$ is arbitrary. Let $C_{\text{slow}}$ be the projection matrix onto the slow eigenspace of $A$ of dimension d. Since we work in the eigenbasis of $A$, $C_{\text{slow}} = [I_d \quad 0]$. At this point, elements of the tangent space take on the particularly simple form

$$\Psi = [M \quad T]$$

where now $M$ is a $d \times d$ skew symmetric matrix and $T \in \mathbb{R}^{d \times (N-d)}$ is arbitrary. A basis for the tangent space is provided by the set of matrices $\{M_{ij}, T_{kl}, i = 2, ...d, j = 1, ..., i-1, k = 1, ..., d, l = 1, ..., N-d\}$ where $M_{ij}$ is a matrix with entry 1 at index $(i, j)$ and $-1$ at index $(j, i)$ and zero otherwise, and $T_{kl}$ is the matrix with entry 1 at index $(k, l)$ and zero otherwise. Denote by $D_\Psi Q_m$ the directional derivative of $Q_m$ along the direction of $\Psi$, viewing $Q_m$ as a function of C (denoted $Q_m[C]$):

$$D_\Psi Q_m = \lim_{\alpha \to 0} \frac{Q_m[C_{\text{slow}} + \alpha \Psi] - Q_m[C_{\text{slow}}]}{\alpha} \qquad (19)$$

Let $\Psi_{ij,kl}$ denote the tangent matrix $[M_{ij} \quad T_{kl}]$. Before calculating $Q_m(C_{\text{slow}} + \alpha \Psi_{ij,kl})$ explicitly, we first observe that as long as the NK iterations are initialized with a diagonal $Q_0$, then the diagonal nature of $C_{\text{slow}}^\top C_{\text{slow}}$ ensures that all $Q_m$ will subsequently remain diagonal matrices. In fact, it

can be shown that $\lim_{k\to\infty} Q_m = Q$ will also be diagonal, in this case. We write $A$ in block form as $\begin{bmatrix} \Lambda_{||} & 0 \\ 0 & \Lambda_{\perp} \end{bmatrix}$, and similarly $Q_{m-1} = \begin{bmatrix} \mathcal{Q}_{||} & 0 \\ 0 & \mathcal{Q}_{\perp} \end{bmatrix}$, where $\Lambda_{||}, \mathcal{Q}_{||}$ are $d \times d$ diagonal matrices defined on the image of $C_{\text{slow}}$ and $\Lambda_{\perp}, \mathcal{Q}_{\perp}$ are diagonal matrices defined on the kernel of $C_{\text{slow}}$. We denote the individual diagonal elements of $\Lambda_{||}, \mathcal{Q}_{||}$ as $\lambda_i, \mathcal{Q}_i, i = 1, ..., d$ and of $\Lambda_{\perp}, \mathcal{Q}_{\perp}$ as $\lambda_i, \mathcal{Q}_i, i = d, ..., N - d$. Then, equation 18 becomes:

$$
\left( \begin{bmatrix} \Lambda_{||} & 0 \\ 0 & \Lambda_{\perp} \end{bmatrix} - \begin{bmatrix} (I_d - \alpha^2 M_{ij}^2)\mathcal{Q}_{||} & (\alpha T_{kl} + \alpha^2 M_{ij}^{\top} T_{kl})\mathcal{Q}_{\perp} \\ (\alpha T_{kl}^{\top} + \alpha^2 T_{kl}^{\top} M_{ij})\mathcal{Q}_{||} & \alpha^2 T_{kl}^{\top} T_{kl} \mathcal{Q}_{\perp} \end{bmatrix} \right) Q_m[C_{\text{slow}} + \Psi_{ij,kl}]
$$

$$
+ Q_m[C_{\text{slow}} + \Psi_{ij,kl}] \left( \begin{bmatrix} \Lambda_{||} & 0 \\ 0 & \Lambda_{\perp} \end{bmatrix} - \begin{bmatrix} \mathcal{Q}_{||}(I_d - \alpha^2 M_{ij}^2) & \mathcal{Q}_{||}(\alpha T_{kl} + \alpha^2 M_{ij}^{\top} T_{kl}) \\ \mathcal{Q}_{\perp}(\alpha T_{kl}^{\top} + \alpha^2 T_{kl}^{\top} M_{ij}) & \mathcal{Q}_{\perp}\alpha^2 T_{kl}^{\top} T_{kl} \end{bmatrix} \right)
$$

$$
+ I_N + \begin{bmatrix} \mathcal{Q}_{||}(I_d - \alpha^2 M_{ij}^2)\mathcal{Q}_{||} & \mathcal{Q}_{||}(\alpha T_{kl} + \alpha^2 M_{ij}^{\top} T_{kl})\mathcal{Q}_{\perp} \\ \mathcal{Q}_{\perp}(\alpha T_{kl}^{\top} + \alpha^2 T_{kl}^{\top} M_{ij})\mathcal{Q}_{||} & \alpha^2 \mathcal{Q}_{\perp} T_{kl}^{\top} T_{kl} \mathcal{Q}_{\perp} \end{bmatrix} = 0 \tag{20}
$$

where we have used $M^{\top} = -M$. The equivalent equation for $Q_m(C_{\text{slow}})$ reads:

$$
\left( \begin{bmatrix} \Lambda_{||} & 0 \\ 0 & \Lambda_{\perp} \end{bmatrix} - \begin{bmatrix} \mathcal{Q}_{||} & 0 \\ 0 & 0 \end{bmatrix} \right) Q_m[C_{\text{slow}}] + Q_m[C_{\text{slow}}] \left( \begin{bmatrix} \Lambda_{||} & 0 \\ 0 & \Lambda_{\perp} \end{bmatrix} - \begin{bmatrix} \mathcal{Q}_{||} & 0 \\ 0 & 0 \end{bmatrix} \right) + I_N + \tag{21}
$$

$$
\begin{bmatrix} \mathcal{Q}_{||}^2 & 0 \\ 0 & 0 \end{bmatrix} = 0 \tag{22}
$$

This latter equation is easily solved to yield:

$$
Q_m[C_{\text{slow}}] = \begin{bmatrix} \frac{1}{2}\left(I_d + \mathcal{Q}_{||}^2\right)\left(\mathcal{Q}_{||} - \Lambda_{||}\right)^{-1} & 0 \\ 0 & -\frac{1}{2}\Lambda_{\perp}^{-1} \end{bmatrix}
$$

To explicitly solve the former equation, we recall that the matrices $M_{ij}$ and $T_{kl}$ have only two and one nonzero terms, respectively. $M_{ij}^2$ contains two nonzero terms at index $(i, i)$ and $(j, j)$. $T_{kl}^{\top} T_{kl}$ contains one non-zero term at index $(l, l)$. $M_{ij}^{\top} T_{kl}$ contains a single nonzero term at $(i, l)$ or $(j, l)$ only if $k = i$ or $k = j$, respectively. Accordingly, we distinguish between where $k = i$ or $k = j$ (without loss of generality we may assume that $k = j$), and where $k \neq i$ and $k \neq j$.

In what follows, we will denote the $(i, j)$ entry of $Q_m[C_{\text{slow}} + \alpha\Psi_{ij,kl}]$ as $q_{ij}$.

1. *Case 1: $k = j$* In this case, careful inspection of eq. 20 reveals that it differs from eq. 22 only within a $3 \times 3$ subsystem:

$$
\begin{bmatrix} \mathcal{S}_{11} & \mathcal{S}_{12} & \mathcal{S}_{13} \\ \mathcal{S}_{21} & \mathcal{S}_{22} & \mathcal{S}_{23} \\ \mathcal{S}_{31} & \mathcal{S}_{32} & \mathcal{S}_{33} \end{bmatrix} = 0
$$

Note that this matrix is symmetric, yielding 6 equations for 6 unknowns:

$\mathcal{S}_{11} = \alpha^2 \mathcal{Q}_i^2 + 2\alpha^2 \mathcal{Q}_{d+l} q_{i,d+l} + \mathcal{Q}_i^2 + 2q_{ii}\left(-\alpha^2 \mathcal{Q}_i + \lambda_i - \mathcal{Q}_i\right) + 1$

$\mathcal{S}_{12} = \alpha^2 \mathcal{Q}_{d+l} q_{j,d+l} - \alpha \mathcal{Q}_{d+l} q_{i,d+l} + q_{ij}\left(-\alpha^2 \mathcal{Q}_i + \lambda_i - \mathcal{Q}_i\right) + q_{ij}\left(-\alpha^2 \mathcal{Q}_j + \lambda_j - \mathcal{Q}_j\right)$

$\mathcal{S}_{13} = -\alpha^2 \mathcal{Q}_i \mathcal{Q}_{d+l} + \alpha^2 \mathcal{Q}_i q_{ii} + \alpha^2 \mathcal{Q}_{d+l} q_{d+l} - \alpha \mathcal{Q}_j q_{ij} +$
$\quad q_{i,d+l}\left(-\alpha^2 \mathcal{Q}_{d+l} + \lambda_{d+l}\right) + q_{i,d+l}\left(-\alpha^2 \mathcal{Q}_i + \lambda_i - \mathcal{Q}_i\right)$

$\mathcal{S}_{22} = \alpha^2 \mathcal{Q}_j^2 - 2\alpha \mathcal{Q}_{d+l} q_{j,d+l} + \mathcal{Q}_j^2 + 2q_{jj}\left(-\alpha^2 \mathcal{Q}_j + \lambda_j - \mathcal{Q}_j\right) + 1$

$\mathcal{S}_{23} = \alpha^2 \mathcal{Q}_i q_{ij} + \alpha \mathcal{Q}_j \mathcal{Q}_{d+l} - \alpha \mathcal{Q}_j q_{jj} - \alpha \mathcal{Q}_{d+l} q_{d+l,d+l} + q_{j,d+l}\left(-\alpha^2 \mathcal{Q}_{d+l} + \lambda_{d+l}\right) +$
$\quad q_{j,d+l}\left(-\alpha^2 \mathcal{Q}_j + \lambda_j - \mathcal{Q}_j\right)$

$\mathcal{S}_{33} = 2\alpha^2 \mathcal{Q}_i q_{i,d+l} + \alpha^2 \mathcal{Q}_{d+l}^2 - 2\alpha \mathcal{Q}_j q_{j,d+l} + 2q_{d+l,d+l}\left(-\alpha^2 \mathcal{Q}_{d+l} + \lambda_{d+l}\right) + 1$

Direct solution is still infeasible, but noting our interest is in the behavior of solutions as $\alpha \to 0$, and only terms of $O(\alpha)$ will survive in the limit in eq. 19, we consider solving these equations perturbatively. That is, we express each $q_{ij}$ in a power series in $\alpha$: $q_{ij} = q_{ij}^{(0)} + q_{ij}^{(1)}\alpha + O(\alpha^2)$. One obtains each coefficient in the expansion by plugging this form into the above matrix and setting all terms of the corresponding order in $\alpha$ to 0. The lowest order term, $q_{ij}^{(0)}$, coincides with the solution of the unperturbed system, eq. 22. Plugging in the expansion into the $3 \times 3$ subsystem above, as well as the solution of the unperturbed system, and collecting all coefficients proportional to $\alpha$ yields the following system of equations:

$$\begin{bmatrix} \mathcal{S}_{11}^{(1)} & \mathcal{S}_{12}^{(1)} & \mathcal{S}_{13}^{(1)} \\ \mathcal{S}_{21}^{(1)} & \mathcal{S}_{22}^{(1)} & \mathcal{S}_{23}^{(1)} \\ \mathcal{S}_{31}^{(1)} & \mathcal{S}_{32}^{(1)} & \mathcal{S}_{33}^{(1)} \end{bmatrix} = 0$$

$$\mathcal{S}_{11}^{(1)} = 2\lambda_i q_{ii}^{(1)} - 2\mathcal{Q}_i q_{ii}^{(1)}$$

$$\mathcal{S}_{12}^{(1)} = \lambda_i q_{ij}^{(1)} + \lambda_j q_{ij}^{(1)} - \mathcal{Q}_i q_{ij}^{(1)} - \mathcal{Q}_j q_{ij}^{(1)}$$

$$\mathcal{S}_{13}^{(1)} = \lambda_i q_{i,d+l}^{(1)} + \lambda_{d+l} q_{i,d+l}^{(1)} - \mathcal{Q}_i q_{i,d+l}^{(1)}$$

$$\mathcal{S}_{22}^{(1)} = 2\lambda_j q_{jj}^{(1)} - 2\mathcal{Q}_j q_{jj}^{(1)}$$

$$\mathcal{S}_{23}^{(1)} = \lambda_j q_{j,d+l}^{(1)} + \lambda_{d+l} q_{j,d+l}^{(1)} + \mathcal{Q}_j \mathcal{Q}_{d+l} - \mathcal{Q}_j q_{j,d+l}^{(1)} - \frac{\mathcal{Q}_j\left(\mathcal{Q}_j^2 + 1\right)}{-2\lambda_j + 2\mathcal{Q}_j} + \frac{\mathcal{Q}_{d+l}}{2\lambda_{d+l}}$$

$$\mathcal{S}_{33}^{(1)} = 2\lambda_{d+l} q_{d+l}^{(1)}$$

Solving this system yields the following solutions for the $q_{ij}^{(1)}$:

$$q_{ii}^{(1)} = 0$$

$$q_{jj}^{(1)} = 0$$

$$q_{d+l,d+l}^{(1)} = 0$$

$$q_{ij}^{(1)} = 0$$

$$q_{i,d+l}^{(1)} = 0$$

$$q_{j,d+l}^{(1)} = \frac{-2\lambda_j\lambda_{d+l}\mathcal{Q}_j\mathcal{Q}_{d+l} - \lambda_j\mathcal{Q}_{d+l} - \lambda_{d+l}\mathcal{Q}_j^3 + 2\lambda_{d+l}\mathcal{Q}_j^2\mathcal{Q}_{d+l} - \lambda_{d+l}\mathcal{Q}_j + \mathcal{Q}_j\mathcal{Q}_{d+l}}{2\lambda_j^2\lambda_{d+l} + 2\lambda_j\lambda_{d+l}^2 - 4\lambda_j\lambda_{d+l}\mathcal{Q}_j - 2\lambda_{d+l}^2\mathcal{Q}_j + 2\lambda_{d+l}\mathcal{Q}_j^2}$$

2. *Case 2*: $k \neq i, k \neq j$. In this case, we must again consider the $3 \times 3$ subsystem indexed by $i, j, d+l$, but since $M_{ij}T_{kl}$ is a matrix of all zeros, the expression simplifies considerably:

$$\begin{bmatrix} \mathcal{S}_{11} & \mathcal{S}_{12} & \mathcal{S}_{13} \\ \mathcal{S}_{21} & \mathcal{S}_{22} & \mathcal{S}_{23} \\ \mathcal{S}_{31} & \mathcal{S}_{32} & \mathcal{S}_{33} \end{bmatrix} = 0$$

$$\mathcal{S}_{11} = \alpha^2\mathcal{Q}_i^2 + \mathcal{Q}_i^2 + 2q_i\left(-\alpha^2\mathcal{Q}_i + \lambda_i - \mathcal{Q}_i\right) + 1$$

$$\mathcal{S}_{12} = q_{ij}\left(-\alpha^2\mathcal{Q}_i + \lambda_i - \mathcal{Q}_i\right) + q_{ij}\left(-\alpha^2\mathcal{Q}_j + \lambda_j - \mathcal{Q}_j\right)$$

$$\mathcal{S}_{13} = \lambda_{d+l}q_{i,d+l} + q_{i,d+l}\left(-\alpha^2\mathcal{Q}_i + \lambda_i - \mathcal{Q}_i\right)$$

$$\mathcal{S}_{22}\alpha^2\mathcal{Q}_j^2 + \mathcal{Q}_j^2 + 2q_j\left(-\alpha^2\mathcal{Q}_j + \lambda_j - \mathcal{Q}_j\right) + 1$$

$$\mathcal{S}_{23} = \lambda_{d+l}q_{j,d+l} + q_{j,d+l}\left(-\alpha^2\mathcal{Q}_j + \lambda_j - \mathcal{Q}_j\right)$$

$$\mathcal{S}_{33} = 2\lambda_{d+l}q_{d+l} + 1$$

Plugging in the power series expansion $q_{ij} = q_{ij}^{(0)} + q_{ij}^{(1)}\alpha + O(\alpha^2)$, one finds the lowest order terms in $\alpha$ within this system of equations occurs at $O(\alpha^2)$, and thus to $O(\alpha)$, the solution of $Q_m[C_{\text{slow}} + \alpha\Psi_{ij,kl}]$ coincides with $Q_m[C_{\text{slow}}]$.

To complete the proof of Theorem 3, we must calculate the following quantity:

$$D_{\Psi_{ij,kl}}\text{Tr}\left(Q_m^2\right) = \lim_{\alpha \to 0} \frac{\text{Tr}(Q_m[C_{\text{slow}+\alpha\Psi_{ij,kl}}]^2) - \text{Tr}(Q_m[C_{\text{slow}}]^2)}{\alpha}$$

From the case-wise analysis above, we see that the only matrix element of $Q_m$ that differs between $Q_m[C_{\text{slow}+\alpha\Psi_{ij,kl}}]$ and $Q_m[C_{\text{slow}}]$ to $O(\alpha)$ is an off-diagonal term $(q_{j,d+l}^{(1)})$. However, this term does not contribute to the trace of $Q_m^2$ at $O(\alpha)$. Thus, we conclude that along a complete basis for the tangent space of $\Omega$ at $C_{\text{slow}}$, $D_{\Psi_{ij,kl}}\text{Tr}\left(Q_m^2\right) = 0$. From this, we conclude that $\nabla_C\text{Tr}(Q_m[C_{\text{slow}}]^2) = 0$ on $\Omega$. The proof of Lemma 2 follows from application of Lemma 3. The proof of Theorem 2 then follows upon combining Lemma 1 and Lemma 2. $\square$

## A.4 Supplementary Figures

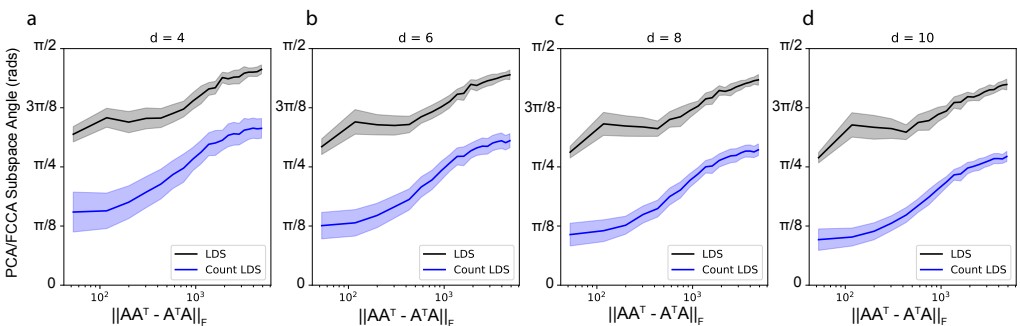

**Figure A1:** (Black) Average subspace angles between FCCA and PCA projections applied to Dale's law constrained linear dynamical systems (LDS) as a function of non-normality. (Blue) Subspace angles between FCCA and PCA projections applied to firing rates derived from spiking activity driven by Dale's Law constrained LDS. Spread around both curves indicates standard deviation taken over 20 random generations of $A$ matrices and 10 random initializations of FCCA. Panels a-d report results at projection dimension $d = 4, 6, 8, 10$, respectively, to complement the results shown in **Figure 2** in the manuscript, demonstrating that non-normality drives the divergence between FCCA and PCA subspaces.

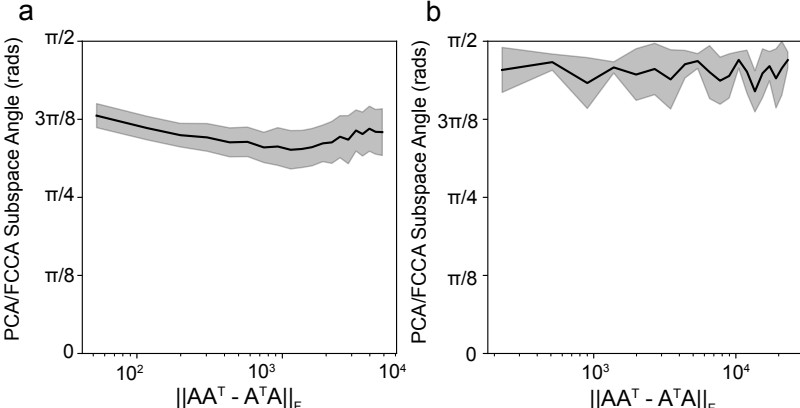

**Figure A2:** Average subspace angles between $d = 2$ FCCA and PCA projections applied to (a) switching linear dynamical system sequence and (b) task optimized RNN as a function of non-normality.

We found that PCA and FCCA identify distinct subspaces in non-normal systems. To evaluate to what degree this observation is robust to non-stationarity and nonlinearity in the data generating process, we simulated data from a switching linear dynamical system and a task optimized RNN (full

details found in section A.1). In **Supplementary Figure A2**, we plot FCCA/PCA subspace angles as a function of non-normality (switching LDS left, task optimized RNN right). We find subspace angles to be consistently large, with only a weak dependence on non-normality. Thus, FCCA and PCA identify distinct subspaces of dynamics in diverse dynamical systems.

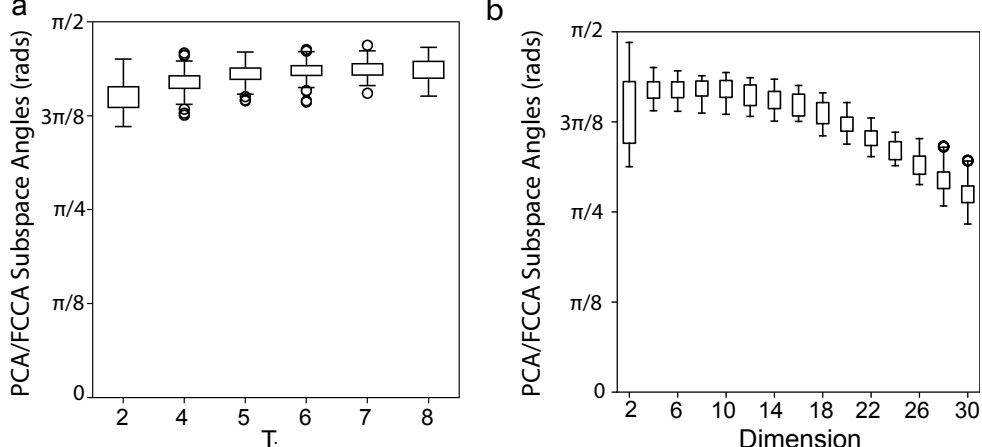

**Figure A3:** (a) Full range of average subspace angles at projection dimension $d = 2$ between PCA and FCCA solutions for various T. Spread is taken over recording sessions and folds of the data within each recording session. (b) Full range of spread in average subspace angles between FCCA for $T = 3$ and PCA taken across 20 initializations of FCCA and all recording sessions.

In **Figure A3**, we investigate the robustness of the substantial subspace angles between FCCA and PCA observed in **Figure 3a** to three sources of potential variability: (i) choice of the $T$ parameter within FCCA, (ii) the dimensionality of projection, and (iii) different initializations of FCCA. In **Supplementary Figure A3 a**, we plot the full range of average subspace angles across recording sessions at projection dimension $d = 2$ between PCA and FCCA for various choices of $T$ ($T = 3$ is shown in **Figure 3a**). We observe that subspace angles remain consistently large ($> 3\pi/8$ rads) across $T$. In **Figure A3b**, we plot the full range of average subspace angles between FCCA (using $T = 3$) and PCA across a range of projection dimensions. The spread in boxplots is taken across both recording sessions and twenty initializations of FCCA. We observe relatively little variability in the average subspace angles for a fixed projection dimensionality. As the projection dimension is increased, we observe the average subspace angles between FCCA and PCA decrease, from $\approx 3\pi/8$ rads to $\approx \pi/4$ rads. This is to be expected, as it is in general less likely that higher dimensional subspaces will lie completely orthogonal to each other. Overall, we conclude that FCCA and PCA subspaces are geometrically distinct in the hippocampal dataset examined.

To evaluate the robustness of FCCA's behavioral predictions to different intializations of the algorithm, we trained linear decoders of rat position from FCCA subspaces obtained from each of twenty initializations of FCCA within each recording session. In **Figure A4**, we plot the full spread in the resulting cross-validated $r^2$ relative to the median cross-validated $r^2$ as a function of projection dimension. By $d = 6$, the range of spread in prediction $r^2$ is less than the corresponding difference between FCCA and PCA $r^2$. We therefore conclude that the behavioral prediction performance of FCCA is robust to the non-convexity of its objective function.

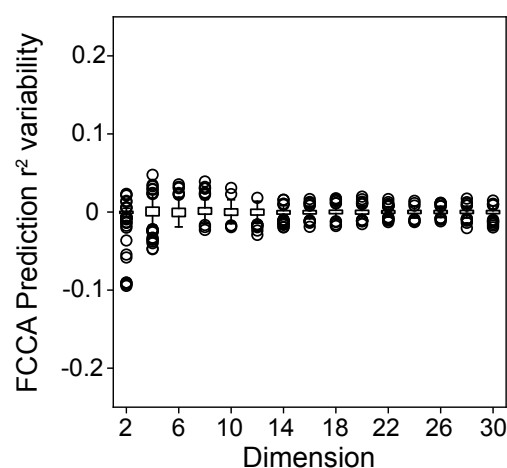

**Figure A4:** Full range of variation in cross-validated position $r^2$ from projected FCCA activity relative to the median cross-validated $r^2$. Spread is taken across 20 initializations of FCCA and across all recording sessions

