# OpenReview forum: "Identifying Feedforward  and Feedback Controllable Subspaces of Neural Population Dynamics"
_ICLR.cc/2025/Conference — ICLR 2025 Conference Withdrawn Submission_

### Official Review · Reviewer_ndQH · 2024-10-31

**Soundness:** 3
**Presentation:** 1
**Contribution:** 3
**Rating:** 3
**Confidence:** 2

**Summary:**

This paper proposes a dimensionality reduction method called **Feedback Controllability Components Analysis (FCCA)**. The method identifies the subspace of a linear dynamical system that is maximizes a measure of feedback controllability, which is defined in terms of the optimal Kalman filter. Specifically, the *feedback controllability* is the trace of the product of two positive definite matrices: one matrix encoding the covariance of the estimation error and another matrix encoding how the optimal control cost depends on initial condition. They compare FCCA with PCA and show that the angle between the subspaces depends on the non-normality of the dynamics matrix. They also apply their method to neural recordings from rat hippocampus, macaque primary motor cortex and primary somatosensory cortex. In each case, they find that the FCCA projection is more predictive of the animal's behavior than the PCA projection.

**Strengths:**

Identifying relevant low-dimensional projections of high-dimensional neural data is a highly relevant topic. This is evidenced by the numerous methods that have been proposed in the literature. This paper proposes a compelling methodology that is based on the controllability of the neural state space. They build on a well-developed theory from the control theory community that is not well-known in the neuroscience community.

**Weaknesses:**

- I found the paper to be dense and required significant effort to understand (at least for me, the main reason for my presentation score of 1 and my overall recommendation for rejection). One of the stated goals of this paper is to highlight important ideas from the control theory community that are relevant to the neuroscience community. I do not think this paper lays out the ideas with sufficient clarity for the theoretical neuroscience community (my confidence score is 2 because I don't think I fully understand the paper). While some of this is perhaps unavoidable due to the material, I think the presentation could be substantially improved with an intuitive figure explaining the concepts in this paper. It would be very helpful to have an illustrative example (e.g. in 2D) of the relationship between $A$, $B$ and the Gramian $\Pi$ and how PCA and feedback controllability subspaces differ. A large portion of the computational neuroscience community is quite familiar with 2D linear dynamical systems and I think this paper misses an opportunity to connect the results to prior understanding in the community.
- FCCA is only compared with PCA in section 4. There are a multitude of methods for extracting subspaces beyond PCA; e.g., slow feature analysis, GFPA, LFADS, etc. Such comparisons seem important if the goal is to encourage practitioners to use FCCA.

**Questions:**

- Is the controllability Gramian $\Pi$ equal to the covariance of the stationary distribution of $x(t)$ defined in eq. (2)? I think this is a more intuitive description that could be stated earlier.
- Notation in section 2.2: on line 182, $Q$ is set equal to the (arg)-min over probability distributions. It appears this means that the optimal distribution is Gaussian with covariance matrix $Q$. Are these minimizations equivalent since the optimal distribution is a mean-zero Gaussian? It might help to state this.
- Line 188: What is the meaning of "$P$ encodes the regulation cost incurred for varying initial conditions"?
- Line 222: What is causal vs acausal Kalman filter? Does acausal mean the Kalman filter applied to the backward process $x_b(t)$?
- Theorem 2: If $A$ is symmetric, then isn't $Re(\lambda(A))=\lambda(A)$? If so, I'd suggest stating this.

---

### Official Review · Reviewer_wdEC · 2024-11-01

**Soundness:** 1
**Presentation:** 1
**Contribution:** 2
**Rating:** 3
**Confidence:** 4

**Summary:**

This manuscript proposes an algorithm, Feedback Controllability Components Analysis (FCCA), to identify the low-dimensional subspace critical for feedback control. The authors claim that FCCA is a dimensionality reduction method that encodes controllability within the data. The authors apply FCCA to both simulations and data analyses.

**Strengths:**

- Authors develop FCCA, which computes the feedback control invariant, $\text{Tr}(PQ)$, directly through the observed neural state $x(t) \in \mathbb{R}^N$.

**Weaknesses:**

1. Throughout the paper, almost every equation lacks either a reference or a derivation, making it difficult for readers to follow. For instance:
* Section 2.1: The Lyapunov equation (3) has no reference. A reference can be [Burl (1998), p.72].
* Section 2.2: The cost functions of the Kalman filter and LQR, the dual Riccati equations (5) and (6), and their associated parameter equations for $Q$ and $P$ lack references. References can be, for example, [Burl (1998), p.243 & p.283] and [Jonckheere & Silverman (1983)].
* Section 2.3: The final FCCA equation (11) has no derivation details. Considering it's the main result of this paper, the derivation should be step-by-step. Additionally, the authors do not provide pseudocode for FCCA, making the algorithm challenging to follow.

---
2. The notation throughout the paper is inconsistent, with numerous typos. For example, in Section 2.2, the term $QC’CQ$ in equation (5) should appear only when the observation model is $y(t) = Cx(t) + v(t)$, where $v(t)$ is Gaussian white noise with covariance $I_d$. Sources such as [Burl (1998), p.243] and [Jonckheere & Silverman (1983)] include $v(t)$, leading their Riccati equations to include $QC’CQ$. However, [Ljung & Kailath (1976)] exclude $v(t)$, and thus their Riccati equation does not include $QC’CQ$. This creates a contradiction:
* The $QC’CQ$ term seems necessary in deriving FCCA in equation (9) because it is dual to the LQR Riccati equation (6).
* However, since the neural state $x(t)$ represents the true observation and $y(t) = Cx(t)$ is only a low-dimensional projection, there should be no noise term $v(t)$.
It is unclear how the authors intend to resolve this contradiction. This further underscores the importance of providing step-by-step derivations from the base model to ensure clarity and avoid such contradictions, making it easier for readers to follow. The authors should clarify their assumptions about the observation model and provide a step-by-step derivation showing how they arrive at equation (5) from their model equation (2), particularly explaining the presence or absence of the $QC'CQ$ term.
---
3. [Kashima (2016)] has shown that the feedforward cost matrix (i.e., the controllability Gramian matrix $\Pi$) is equivalent to the data covariance. Additionally, Section 2.2 on feedback controllability closely follows the results in [Jonckheere & Silverman (1983)], including Theorem 1 and the similarity invariant matrix $QP$. Therefore, the only novel contribution of this paper is the derivation of FCCA in Section 2.3. However, this derivation contains serious typographical errors and lacks sufficient detail.

The first issue with FCCA is that it should be $x_b(t) = \Pi x_a(t)$ rather than $x(t) = \Pi x_a(t)$ in line 239, which would allow the authors to derive the equation in line 241 from equation (8). Although this may seem like a minor typo, it is critical because $x_a(t)$ is no longer connected to $x(t)$. Since $x(t)$ represents the observed neural state rather than a latent state, it is unclear how the covariance and cross-covariance of $x_a(t)$ could be estimated from $x(t)$ as required in equation (11), the FCCA formula. The authors should provide a detailed explanation of how they connect $x_a(t)$ to the observed neural state $x(t)$, and how this connection allows for the estimation of covariances $\tilde{P}$ in equation (11).

---
4. The second issue with FCCA is that the Riccati equation (9) for $\tilde{P}$ is not the same as the Riccati equation (6) for $P$. It is unclear why the authors equate $\text{Tr}(QP)$ with $\text{Tr}(Q\tilde{P})$. A detailed derivation linking these two expressions is necessary. Additionally, it is not explained why equation (9) aligns with the Riccati equation associated with the modified LQR cost function (10). Since $C’C$ is replaced by $\Pi^{-1}BB’\Pi^{-1}$ in (10), the final LQR Riccati equation should exclude $C’C$, which contradicts equation (9). A detailed derivation of this step is also required. The authors should provide a step-by-step derivation showing how equation (9) can be transformed into equation (6), so they're equivalent, and how it relates to the modified LQR cost function in equation (10).

---
5. The third issue with FCCA is that equation (11) lacks a derivation. This should be derived step-by-step, as it represents the main result of the paper. Furthermore, some details in equation (11) appear questionable. For instance, following my previous point that $\Pi x_a(t) = x_b(t)$ rather than $x(t)$, the inverse matrix $\Sigma_T^{-1}(C)$ of $\tilde{P}$ in (11) should be based on $x_b(t)$, not $x(t)$. Therefore, this formula appears to be incorrect. These issues need to be clarified through a detailed derivation. The authors should provide a complete, step-by-step derivation of equation (11), starting from the basic assumptions and clearly stating any approximations or simplifications made along the way.

---
6. Beyond the theoretical issues mentioned, a more fundamental question arises: Why not simply fit the parameters $A$ and $B$ of the linear dynamical model in equation (2) directly and solve the dual Riccati equations (5) and (6) iteratively? This would yield the Gramian matrix $\Pi$, the Kalman error covariance $Q$, and the LQR cost matrix $P$. This approach is straightforward since $x(t)$ is the observed neural signal, not a hidden latent state, making it feasible to fit $A$ and $B$ through simple linear regression. Is there any practical advantage to using FCCA equation (11) compared to this more direct approach? A detailed comparison between the FCCA method and the more direct approach, including computational complexity, accuracy, and any other relevant factors will be helpful.

---
7. The simulations do not validate the correctness of FCCA. To achieve this, the authors should simulate a linear dynamical system (which appears to be the LDS model in Figure 2), compute $Q$ and $P$ using the dual Riccati equations (5) and (6), and find $C$ by minimizing $\text{Tr}(QP)$. Then, they should compute $Q$ and $\tilde{P}$ using equation (11) and confirm that these values match $Q$ and $P$ from equations (5) and (6). Finally, the optimal $C$ derived from (11) must align with the original $C$ that minimizes $\text{Tr}(QP)$. This verification should be straightforward since all system parameters are precisely defined in the simulations. Please perform this validation and include the results in a new figure or table in the paper.

---
8. The simulations also fail to validate the correctness of Theorem 2. The authors should simulate the case where $B = I_N$ and $A = A’$, which would make the angle between the PCA and FCCA subspaces equal to zero. In other words, Figure 2a should include the case where $\|AA'-A'A\|_F=0$.


---
Overall, while the overall idea of finding the subspace with feedback controllability is nice, the derivation of FCCA is incomplete and unconvincing and novelty is unclear compared with prior work. Furthermore, the simulations do not properly validate the method's correctness and the data analyses are limited and unconvincing for demonstrating that the method works.


**References**
Burl, J. B. (1998). Linear optimal control: H (2) and H (Infinity) methods. Addison-Wesley Longman Publishing Co., Inc..

**Questions:**

1. The cost function for LQR (line 173) is ambiguous. If the integral is Riemann, there should be no $\frac{1}{T}$ before the integration [Burl (1998), p.283]. If it is instead an Ito integral, please indicate this and use the notation in [Jonckheere & Silverman (1983)], as this refers not to the limit but to the “limit in the mean.”

---
2. The proof of Theorem 2 (Appendix A.3) is difficult to follow due to the contradictions and typographical errors noted in the weaknesses. Once the above weaknesses are resolved, please review the proof of Theorem 2 carefully to ensure its notations are consistent with the main text. Additionally, provide references for any equations that are sourced from other works but not derived here.

---
3. [Page 7, footnote] The normal matrix can be neither symmetric nor orthogonal. $A$ is a normal matrix iff $AA’ = A’A$. One example is
$$
A=\left[\begin{array}{ccc}
1 &1 & 0 \\
0 & 1 & 1 \\
1 & 0 & 1
\end{array}\right]
\Rightarrow
AA' = A'A = \left[\begin{array}{ccc}
2 &1 & 1 \\
1 & 2 & 1 \\
1 & 1 & 2
\end{array}\right]
$$
Clearly, $A$ is neither symmetric nor orthogonal. Please fix this terminology throughout the paper.

---
4. A complete set of figures for other data types (M1 random, S1 random, and M1 maze) should be included in the appendix. Each dataset should have a figure similar to Figure 3, in addition to the information in Table 1. This is crucial to demonstrate that FCCA is applicable across multiple datasets, especially as this is the only data analysis presented in the paper.

---
**Minor problems:**
1. [Line 129] In equation (3), the dummy variable $dt$ should be after $e^{At}BB'e^{A't}$.
2. [Line 141] The volume of reachable state space is proportional to $\sqrt{\det(\Pi)}$, not $\det(\Pi)$, since the volume formula of an $n$-dimensional ellipsoid is $\frac{\pi^{n/2}}{\Gamma(n/2+1)} \prod_{k=1}^{n} r_k$ where $r_{1:n}$ are radius of the ellipsoid.
3. [Line 479] This is a typo. It should be Fig. 3b. There is no Fig. 3c.
4. [Line 325-326] “It can be shown that… to the truncated system…” Please provide references for this statement since it can be shown.

---

### Official Review · Reviewer_2qj1 · 2024-11-04

**Soundness:** 2
**Presentation:** 3
**Contribution:** 2
**Rating:** 5
**Confidence:** 3

**Summary:**

This study proposes a method to identify feedback controllable subspaces from neural population responses. It shows the feedback control can identify different subspaces with feedforward control, and can explain behavioral output better.

**Strengths:**

1. It seems to be novel to apply the feedback controllable subspace to analyze neural data, while the theory of feedback control (subspace) theory has been well grounded.

2. The writing of the paper is good and structure-wise. The math derivations are well laid out.

**Weaknesses:**

1. The data analysis seems a bit preliminary. I suggest the author could identify more intricate details of the identified subspaces from feedforward and feedback controls, and provide physical and behavioral interpretations of these subspaces if possible.

2. The equation just below Eq. 6: should the last $x$ be $x_0$?

**Questions:**

I have no problem with the math and methods developed in this study, but I have a major conceptual question about feedback control.

Conceptually, feedback control means the controller sends feedback signals back to the system (neural circuits) based on the system's outputs. Conventionally, people think the behavioral output is a (feedforward) readout of the internal state of neural circuits. Then a gap is why behavioral outputs need to send feedback to neural circuits. Further, why do behavioral outputs need even to identify the feedback controllable subspaces? Does this operation complicate the information processing or bring some actual benefits to the brain?
I can accept that the motor system and motor output (behavior) need feedback control, but I am still struggling to understand why the sensory cortex also needs that (S1 random in Table 1). I think some conceptual explanations about this are quite helpful, rather than just using this as a data analysis method to show its improved performance.

---

### Official Review · Reviewer_fhBt · 2024-11-09

**Soundness:** 4
**Presentation:** 3
**Contribution:** 4
**Rating:** 8
**Confidence:** 4

**Summary:**

This paper provides a very interesting way of thinking about relevant subspaces in recorded data. Firstly, the paper relates PCA to a specific form of feedforward control, and shows that PCA recovers the dimensions in the neural data that are most affected by an external control input. Secondly, the paper identifies a low-dimensional controller to control recorded dynamics, and defines the most controllable dimensions as the 'Feedback Controllability' components. This paper provides a conceptual advance to the field and further shows that non-normal dynamics that arise in constrained networks lead to orthogonal PCA and FCCA components, and that the feedback controllable components provide a good reconstruction of behavior.

**Strengths:**

The paper is well-motivated and well-reasoned. The findings are very interesting and highly relevant to the neuroscience and data analysis communities. The conceptual advances are very high.

**Weaknesses:**

The assumptions that the input is purely white noise is potentially problematic. Could the authors show, at least in simulation, that their main results hold with temporally filtered signals?

While Theorem 1 is helpful for the paper, it may not be necessary to restate it in its entirety, or might be sufficient in the Appendix.

Some key references may be missing, such as to the Henrici metric for non-normality.

**Questions:**

How would Figure 3b look if compared with reduced rank regression with the relevant number of dimensions? i.e., what is the best decodability with a limited number of dimensions? To follow up on this further, are the dimensions identified by reduced rank regression also somewhat orthogonal to those identified by PCA?

---

### Note · Authors · 2024-11-20

**Comment:**

We are very thankful for these reviews, and in particular thank reviewer wdEC for their in depth technical comments. We agree with all points of all reviewers and have versions of the manuscript that address them, and further believe all technical issues can be clarified/justified. However, that manuscript is too long for the ICLR format and contains interesting neuroscience results that are most appropriate for a neuroscience audience. Thus, given the diversity of requests (more neuroscience, more control theory intuition, more detailed analytics) we have elected to not pursue further consideration at ICLR.

**Withdrawal Confirmation:**

I have read and agree with the venue's withdrawal policy on behalf of myself and my co-authors.